# ORFanID: A web-based search engine for the discovery and identification of orphan and taxonomically restricted genes

Richard S. Gunasekera[1]*, Komal K. B. Raja[2], Suresh Hewapathirana[3], Emanuel Tundrea[4], Vinodh Gunasekera[5], Thushara Galbadage[6], Paul A. Nelson[7]

1 Department of Chemistry, Physics and Engineering, School of Science, Technology & Health, Biola University, La Mirada, CA, United States of America, 2 Department of Pathology & Immunology, Baylor College of Medicine, Houston, TX, United States of America, 3 European Bioinformatics Institute, Welcome Genome Campus, Hinxton, Cambridgeshire, United Kingdom, 4 Griffiths School of Management and IT, Emanuel University of Oradea, Oradea, Romania, 5 Bioinformatics, Chesalon USA, Inc., Houston, TX, United States of America, 6 Department of Kinesiology and Public Health, School of Science, Technology & Health, Biola University, La Mirada, CA, United States of America, 7 Biola University, La Mirada, CA, United States of America

* richard.gunasekera@biola.edu

**Data Availability Statement:** The ORFanID Web Application and Database are freely available to non-commercial users at http://www.orfangenes.

## Abstract

With the numerous genomes sequenced today, it has been revealed that a noteworthy percentage of genes in a given taxon of organisms in the phylogenetic tree of life do not have orthologous sequences in other taxa. These sequences are commonly referred to as "orphans" or "ORFans" if found as single occurrences in a single species or as "taxonomically restricted genes" (TRGs) when found at higher taxonomic levels. Quantitative and collective studies of these genes are necessary for understanding their biological origins. However, the current software for identifying orphan genes is limited in its functionality, database search range, and very complex algorithmically. Thus, researchers studying orphan genes must harvest their data from many disparate sources. ORFanID is a graphical web-based search engine that facilitates the efficient identification of both orphan genes and TRGs at all taxonomic levels, from DNA or amino acid sequences in the NCBI database cluster and other large bioinformatics repositories. The software allows users to identify genes that are unique to any taxonomic rank, from species to domain, using NCBI systematic classifiers. It provides control over NCBI database search parameters, and the results are presented in a spreadsheet as well as a graphical display. The tables in the software are sortable, and results can be filtered using the fuzzy search functionality. The visual presentation can be expanded and collapsed by the taxonomic tree to its various branches. Example results from searches on five species and gene expression data from specific orphan genes are provided in the Supplementary Information.

com/. The Development Stack of ORFanID comprises a VueJs front-end, a back-end in the Spring Boot framework, and databases in PostgreSQL. The software modules of ORFanID are containerized using Docker and communicate using RESTful APIs. ORFanID supports all major browsers. The Source Code for ORFanID is freely available at https://github.com/orfanid

**Funding:** This work was supported in part by the Roth Family- received by RG. The funders did not and will not have a role in study design, data collection and analysis, decision to publish, or preparation of the manuscript.

**Competing interests:** The authors have declared that no competing interests exist.

## 1. Introduction

Following the introduction of large-scale, high-throughput automated DNA sequencing in the mid-1990s, the comparative analysis of whole genomes revealed a large number of protein-coding open reading frames (ORFs) that occur only in one species or as taxonomically restricted genes (TRGs). These TRGs occur at systematic ranks from the genus upwards. ORFs that are specific to one species are designated as "orphan" (sometimes spelled "ORFan") genes, while more widely distributed ORFs that are not present universally but only at lower taxonomic ranks, are designated as TRGs [1]. By definition, orphan genes lack detectable homology in other species, and TRGs are not present in any genomes outside their respective taxa. For example, a TRG may be found only in the genus *Drosophila*, but not in any other Dipteran or any larger systematic category such as Insecta, Arthropoda, and so on.

Early molecular genetic studies focused on conserved genes, either common to all organisms (e.g., ribosomal genes) or broader systematic categories (e.g., the Wnt signaling pathway in Metazoa), as it was difficult to study TRGs or orphans when relying on the amplification and sequencing of sub-kilobase regions. The neglect of orphans has continued to some extent in the genomic era, as functional roles are mainly assigned to newly sequenced genes mainly via homology criteria (e.g., existing annotations in other species). However, it has been shown in some cases that orphan genes are uniquely involved in making one species distinct from another phenotypically [2–4]. The last decade has seen a rapidly increasing interest in studying orphan genes [5–7]. Genomic sequencing has revealed that large fractions of genes in a given species' complete genome do not possess orthologous sequences in other species, so TRGs represent important mediators of phenotypic novelty [8–10].

The prevailing theory, given the monopoly of life found on Earth, posits that every gene in existence today descended from genes present in the Last Universal Common Ancestor (LUCA). This implies that the genes we see today are directly descended from sequences found in LUCA. Various molecular mechanisms; for example, gene duplication, recombination, and divergence have given rise to novel protein-coding regions (or open reading frames) in the genome, and their associated cell functions [11–13].

More significantly, these origination processes are not expected to eliminate historical traces of their origins, but rather to continue to show these homologies according to prevailing theory. Essentially, this means that we should see a connection between the core set of sequences seen in LUCA and all the genes stemming from this early ancestor.

However, the discovery of orphan genes has led to a changing picture of gene evolution and formation suggesting that de novo formation may be a predominant mechanism for gene emergence [10]. This shift in thinking has significant ramifications for understanding genes, the genome's non-coding regions, and fully functional sequences and calls for the re-analysis of existing data and comparative genomic analysis to provide new and accurate understandings [14, 15]. Several mechanisms have been proposed to explain the generation of orphans/TRGs, including de novo gene birth, divergence beyond recognition, and horizontal gene transfer [16]. It should be noted that ORFanID does not distinguish between these possibilities. Its role is to identify orphan/TRGs based on detectable homology. The interpretation of the underlying mechanism that led to these orphan/TRGs is up to the user, based on the specific context and supporting evidence. Therefore, users should be aware of these different possibilities as they analyze their results. These represent current understanding, although additional mechanisms may be discovered or proposed in future research. The biological understanding of the origins and the functions of orphan genes will have applications in medicine and evolutionary biology across the tree of life. Therefore, orphan genes (and TRGs) represent an

intriguing aspect of biology, lying at the intersection of genomics, genetics, comparative and structural biology, phylogenetics, and evolution.

To study growing numbers of these novel genes across genomes, easy-to-use bioinformatics tools are needed that can be utilized by scientists from the life sciences and those with computational backgrounds. Tools such as BLAST and related software have been made available [1, 16, 17]. However, further tools are necessary for discovering orphan genes and assigning TRGs to their taxonomic levels (sometimes known as phylostratigraphy) [18].

Currently, the tool options researchers can find today for studying orphan genes are limited, as most software solutions focus on identifying orthologs or inferencing ortho groups and are generally limited to proteins. For instance, ORFanFinder functionality is limited to plants, bacteria, and fungi, and the URL in the original publication (DOI: 10.1093/bioinformatics/btw122) is not active [17]. However, a web search has revealed that ORFanFinder is available at http://bcb.unl.edu/orfanfinder/ although this software has not been updated since 2016. SequenceServer performs BLAST without classifying the proteins/DNA sequences into taxonomic levels [19]. The software Geneious can perform alignment and build a phylogenetic tree but identifying orphans using this software is challenging [20]. Similarly, OrthoFinder provides the option to use DIAMOND or its recommended MMseq2 for sequence alignment [21–24]. OMA orthology or the series of analytical resources developed by the Bioinformatics Resource Centers (BRCs) for Infectious Diseases program [bioinformatics tools, workspaces, and services for bioinformatics data analysis like AmoebaDB, FungiDB, OrthoMCL] only show orthologous genes/proteins and do not identify orphan genes [25, 26]. A newer gene classification platform, www.shoot.bio, may also be helpful to align and compare gene origins but it only uses protein (amino acid) sequences. In short, these tools perform alignment or identify conserved genes from genomes, but they are not suited to identify orphans as ORFanID is designed to do.

ORFanID's distinctiveness is in three aspects: (1) It processes not just protein/amino acid sequences but also DNA/nucleotide sequences. (2) With its built-in homology interpreter and classifier, this search engine provides the taxonomic rank of a gene either as an orphan gene or as a gene restricted to a taxonomic level in the tree of life; (3) As ORFans and TRGs are identified, ORFanID builds its own database with the results of the analysis and provides the researcher with the possibility to further explore the data.

## 2. Methods

### 2.1 Algorithm and implementation

ORFanID identifies orphan genes and TRGs from a given list of DNA or protein sequences mainly using NCBI accession numbers (Fig 1). It detects homologous sequences in the NCBI non-redundant databases by using the BLAST alignment tool. All the previously mentioned tools acknowledge that NCBI databases are among the most trusted sources. BLAST is a well-established tool, but its slowness in processing is acknowledged as the databases continue to grow rapidly. Since the ORFanID search engine utilizes local BLASting and a 24-core 48-thread server which significantly improves the execution speed of BLAST as compared with the web-based BLAST searches.

If the query sequence is a protein, the BLASTP algorithm is used, while the BLASTN algorithm is used for the nucleotide sequences. In addition, ORFanID also supports PSI-BLAST which uses position-specific scoring matrices to detect distant evolutionary relationships between sequences. ORFanID allows the user to submit multiple sequences in FASTA format, which are then processed using software message brokering techniques. The results are combined into a single blast report in a tab-delimited format that can be downloaded by the user.

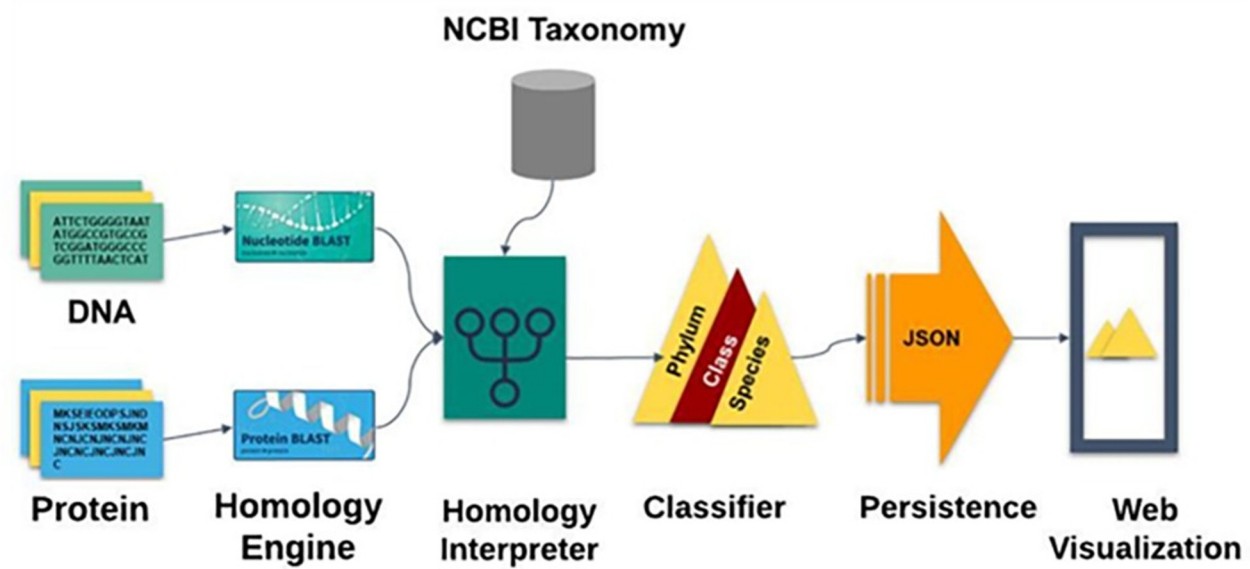

**Fig 1. A graphical representation of the core engine of ORFanID.** DNA and protein sequences are fed through a homology interpreter and classified. ORFanID is a web-based application developed on the Java Spring Framework.

ORFanID sends a customized BLAST command to retrieve the taxonomy IDs of each hit. Using the rank lineage information file provided by the NCBI Taxonomy Database, the ORFanID Homology Interpreter determines the defined taxonomy level (species, genus, family, order, class, phylum, kingdom, and superkingdom) for each sequence found in the BLAST results (Fig 2).

After finding the complete lineage (Linnaeus taxonomic category) for each individual homolog of the input DNA or protein sequence, the ORFanID Classifier starts searching from the Superkingdom level and works its way toward the Species level to find additional homologs (Fig 3). The taxonomic rank where the homologs are located designates them as a "Taxonomically Restricted Gene" (TRG) cluster of orthologs (homologs) by the use of the classifier. ORFanID uses the single ortholog de-classifier rule, meaning that it is sufficient for finding only one ortholog at a certain level in the tree of life, to classify the gene of interest as a TRG at that level or as an orphan at the final level. ORFanID does not require a significant number of orthologs (as the SMOTE technique would) to determine that a sequence is not an orphan [27]. If no orthologous sequences are found, and the ORF remains a species-unique sequence, it will be classified as an "Orphan Gene." If the ORF is unique at the subspecies level, the gene is classified as a "Strict Orphan". Using this algorithm, ORFanID identifies and displays both orphan genes and TRGs. The results can be viewed and analyzed graphically. Further details on the ORFanID algorithm are shown in Fig 4.

### 2.2 Operation

ORFanID accepts either protein or nucleotide (gene) sequences singly or as multiple gene sequences in the FASTA format (Fig 5). Users can easily retrieve multiple protein or gene sequences by providing multiple accessions to the sequence search engine. ORFanID supports both NCBI as well as Uniprot accessions. Optionally, the user can upload a FASTA file or directly copy the sequence into the ORFanID engine as specified on the ORFanID website. Next, users can select a species from the dropdown menu, which includes species' scientific names, NCBI taxonomy IDs, and images of the species for easy visual recognition. Finally, the

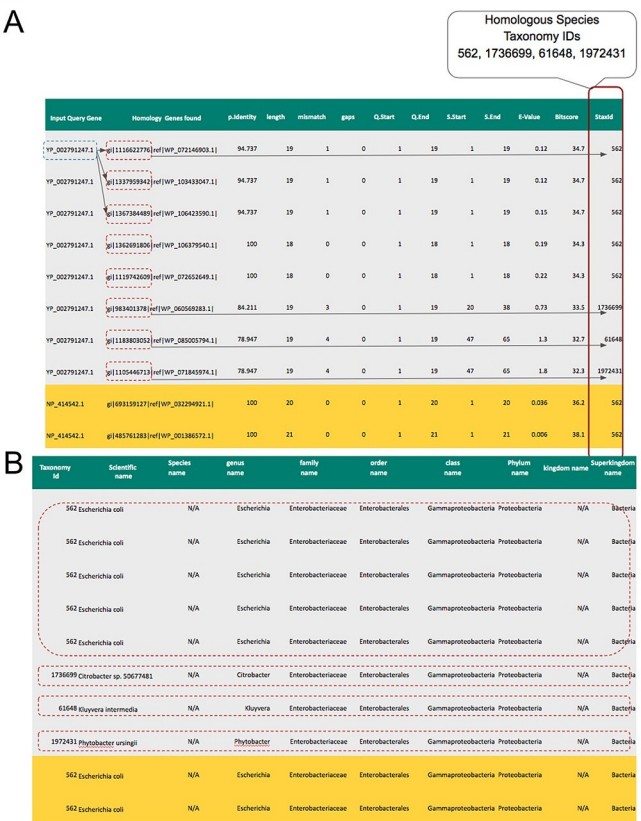

**Fig 2. Homologies and taxid found by ORFanID for the input sequence.** (A) Displays the homologies and taxid found by ORFanID for the input sequence. In this example, the homologies and taxids of two sequences, YP_002791247.1(Q1) and NP_414542.1(Q2), [i.e., genes from Escherichia Coli species (NCBI Taxonomy ID: 562)]. Eight homologous hits were found from four species consisting of the taxonomy ids: 562, 1736699, 61648, and 1972431. (B) Shows the traversal of the algorithm based on the taxid of the sequence homologies that are found by ORFanID. In this example, eight hits that are homologous to the YP_002791247.1 gene and two hits that are homologous to NP_414542.1 are shown. Then the rank lineages are extracted from the taxonomy lineage file in NCBI for each hit.

accuracy of the results can be fine-tuned using advanced parameters such as E-value, percent identity, and a maximum number of target sequences for each BLAST and PSI-BLAST search (Fig 6). The default values for the advanced parameters are; max target sequences, 500; percent identity, 40%; E-value threshold, $10^{-3}$; as used in most cited papers. The threshold for identifying non-orphans could be adjusted for more accuracy, especially in comparative studies. For example, if the goal is to compare genes in a narrow taxonomic range to those of a broader taxonomic range, then a stricter threshold should probably be used.

ORFanID has a dedicated web page that outlines the installation and operating instructions and includes video tutorials to help users understand its web interface. As previously noted, the sequence submissions page also provides four example sets of input gene data for demonstrational purposes, which can be input either as FASTA file sequences or NCBI accession numbers. These four examples analyze sequences of the following species: *Escherichia coli* (562), *Drosophila* melanogaster (7227), *Homo sapiens* (9606), and *Arabidopsis thaliana* (3702). The ORFanID application is designed for ease of use, rich graphics, and reasonable speed to provide a research tool for identifying orphan genes at all taxonomic levels.

| Taxonomy Id | Scientific name | Species name | genus name | family name | order name | class name | Phylum name | kingdom name | Superkingdom name |
|---|---|---|---|---|---|---|---|---|---|
| 562 | Escherichia coli | | Escherichia | Enterobacteriaceae | Enterobacterales | Gammaproteobacteria | Proteobacteria | | Bacteria |
| 1736699 | Citrobacter sp. 50677481 | | Citrobacter | Enterobacteriaceae | Enterobacterales | Gammaproteobacteria | Proteobacteria | | Bacteria |
| 61648 | Kluyvera intermedia | | Kluyvera | Enterobacteriaceae | Enterobacterales | Gammaproteobacteria | Proteobacteria | | Bacteria |
| 1972431 | Phytobacter ursingii | | Phytobacter | | Enterobacterales | Gammaproteobacteria | Proteobacteria | | Bacteria |

| Taxonomy Id | Scientific name | Species name | genus name | family name | order name | class name | Phylum name | kingdom name | Superkingdom name |
|---|---|---|---|---|---|---|---|---|---|
| 562 | Escherichia coli | N/A | Escherichia | Enterobacteriaceae | Enterobacterales | Gammaproteobacteria | Proteobacteria | N/A | Bacteria |

**Fig 3. Gene classification on ORFanID.** The search starts from the superkingdom and steps through each subsequent taxonomy to the species level until it finds homologous sequences. If no homologous sequences are found, the sequence is classified as an ORFan gene.

The protocol described in this peer-reviewed article is published on protocols.io (doi.org/10.17504/protocols.io.14egn37jql5d/v1) and is included for printing purposes as S1 File.

## 3. Results and evaluations

We tested the functionality of ORFanID by analyzing the protein and DNA sequences (or NCBI accessions, version 1.2 on November 17th, 2022) of various organisms, such as *C. elegans*, *S. cerevisiae*, *D. melanogaster*, and *H. sapiens* (Tables 1 and S1). We adjusted the parameters (e-value $< = 10^3$) to filter low-quality results (transposable elements, low-complexity protein regions, etc.) from our analyses. Our results show that ORFanID effectively assigns the protein accessions to their respective taxonomic levels (Tables 1 and S1). For instance, paired box-containing Pax-6 proteins are tissue-specific transcriptional factors highly conserved across most animal phyla. As expected, ORFanID classified *Drosophila* melanogaster Pax-6 proteins, Eyeless (NP_726607.1), and Twin of eyeless (NP_001259080.1) phylum and class-level proteins, respectively.

Similarly, we tested several proteins in *Homo sapiens*, including N-cym (NP_001316897.1) and SPANX (NP_073152.2). The N-cym protein regulates the stability of the proto-oncogene MYCN in neuroblastoma cells, while the SPANX family proteins are expressed in human spermatozoa and are extensively studied in Down syndrome patients [28]. A previous study predicted that these two proteins were restricted to the order Primates [29]. Surprisingly, ORFanID showed that only N-cym is order restricted, while SPANX was classified as a family-restricted protein. To understand this disparity, we searched OrthoDB and the non-redundant database of NCBI using *H. sapiens* SPANX protein (NP_073152.2) as the query [30].

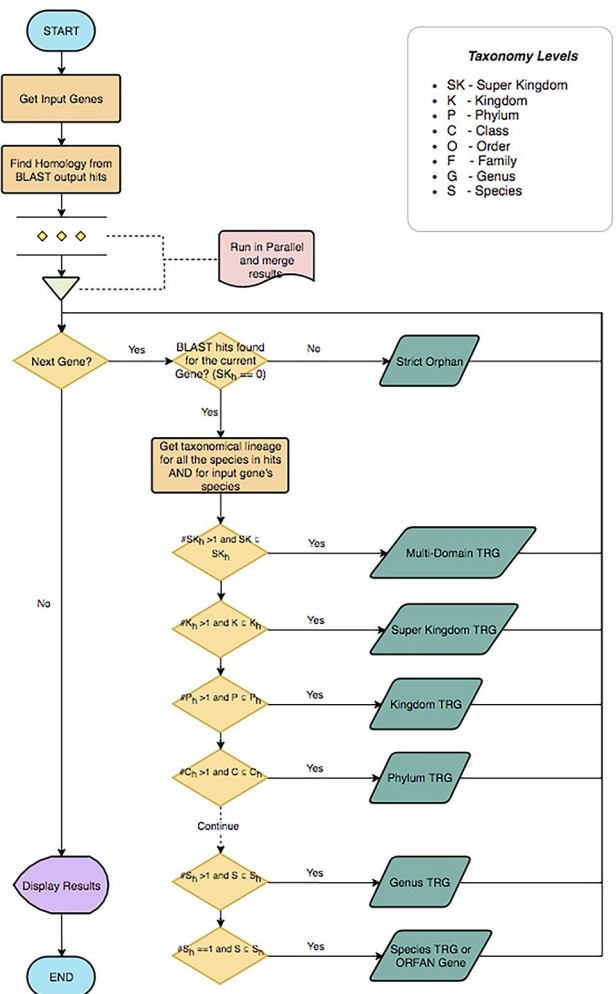

**Fig 4. An example of the algorithm of the ORFanID engine is explained in a UML flow chart diagram.** To find the homolog, our algorithm begins its scan at the Super Kingdom rank. Let $SK_h$ be the set of Super Kingdom taxonomies of the subject genome. If $\#SK_h > 1$, there are no common homologs to be found. This gene (G) will be categorized as a "*Multi-domain*" Taxonomically Restricted Gene (TRG). However, if all of the subject genomes belong to the same Super Kingdom as the input genome ($\#SK_h = 1$), the algorithm moves to the next level which is the Kingdom (K) rank. Let $K_h$ be the set of Kingdom taxonomies of the homologous genomes. If $\#K_h > 1$ the subject sequences belong to multiple Kingdoms. This means that while the Kingdoms are not common, their Super Kingdom is common to the subject genomes. Therefore, if the homolog can be found at the Super Kingdom rank, the gene (G) will be classified as "*Super Kingdom*" TRG. If all the subject genomes belong to the same Kingdom ($\#K_h = 1$), the algorithm moves to the Phylum level. Let $P_h$ be the set of Phylum taxonomies of the subject genomes. ORFanID then checks to see how many distinct Phyla ($\#P_h$) are found in the subject sequences. If $\#P_h > 1$ Phyla are not common, but their Kingdom is common. If the homolog can be found at Kingdom ranks, therefore the gene (G) will be categorized as a "*Kingdom*" TRG. In this way, this search will continue to the Class(C), Order(O), Family(F), Genus(G), and Species(S) levels. If a homolog is found, the ORFanID algorithm will identify the sequence as a TRG at the appropriate taxonomical rank. If the homologs are found only at the Species level the input sequence is identified as an "ORFan Gene". If there are no homologs even at the Species level, the sequence is named a "Strict Orfan".

Interestingly, both databases show that SPANX proteins are restricted to the Hominidae family. These results support that the ORFanID algorithm is accurate in classifying proteins to their respective taxonomic group based on the functionality of BLAST and the choice of parameter settings.

Next, to accurately identify species-specific orphan genes, we tested genes from various species that were previously shown to be orphans by using ORFanID. The *Arabidopsis thaliana*

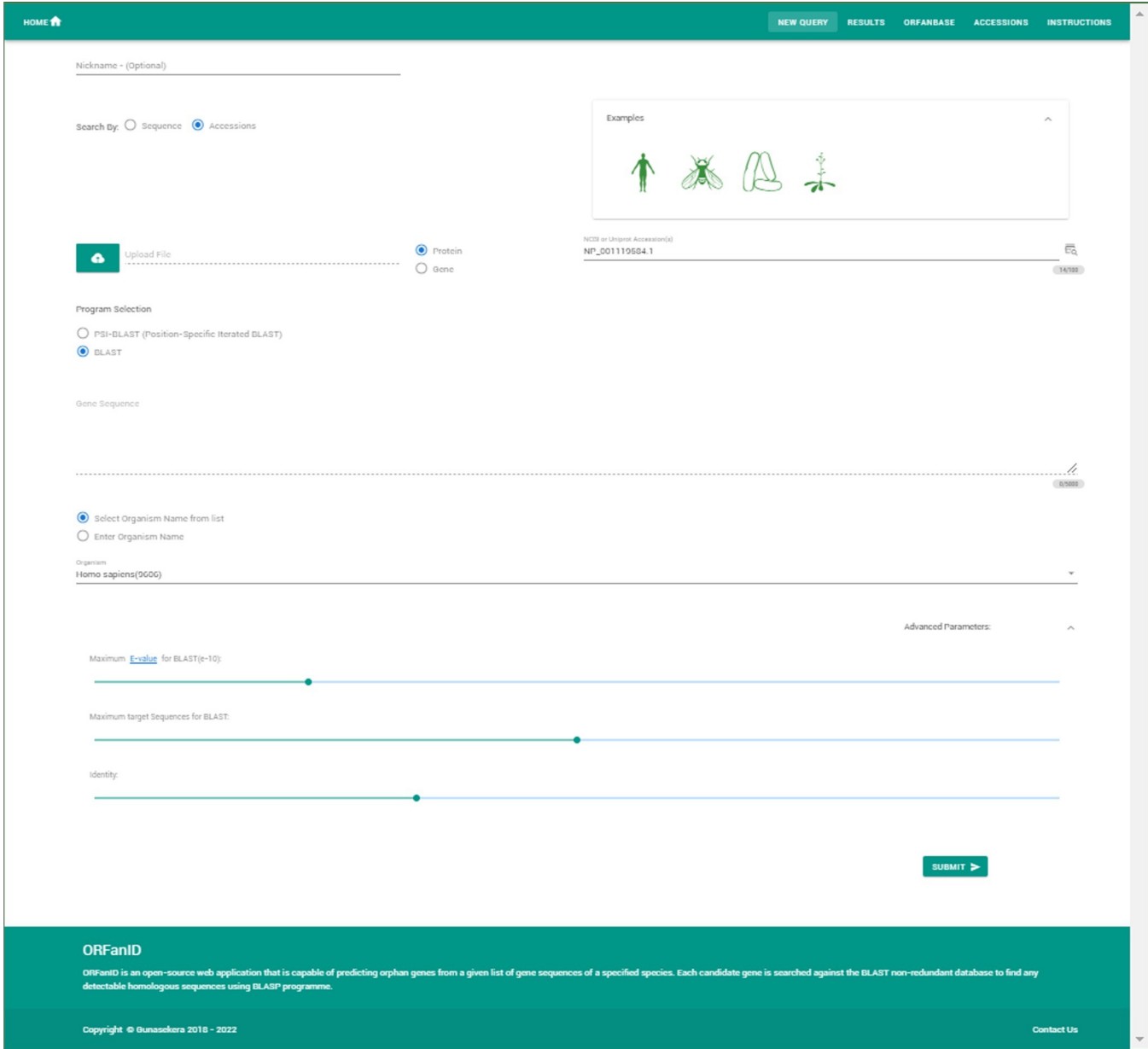

**Fig 5. ORFanID input page.** Four distinct examples are provided to new users for demonstrative purposes. After the user submits their sequence, species, and advanced parameters, ORFanID will perform the steps discussed in the Algorithm section above (Fig 4) and display results as in Fig 6A.

(*A. thaliana*) *QQS* gene was one of the first plant genes shown to be a species-level orphan [31]. It regulates starch biosynthesis in leaves, increases seed protein, and enhances resistance to pathogens [31, 32]. Our results show that ORFanID accurately classifies QQS protein (NP_189695.1) as an *A. thaliana*-specific protein. Similarly, we tested the species-specific orphan genes of some model organisms, such as *D. melanogaster*, *C. elegans*, and *S. cerevisiae*. ORFanID precisely identified these published genes as species-specific orphans. The *D. melanogaster* group-specific orphan genes, *jeanbaptiste*, and *karr* are predominantly expressed in the male germline and are functionally important [33]. RNAi-mediated knock-down of these genes leads to male-specific developmental defects and partial lethality. These genes were

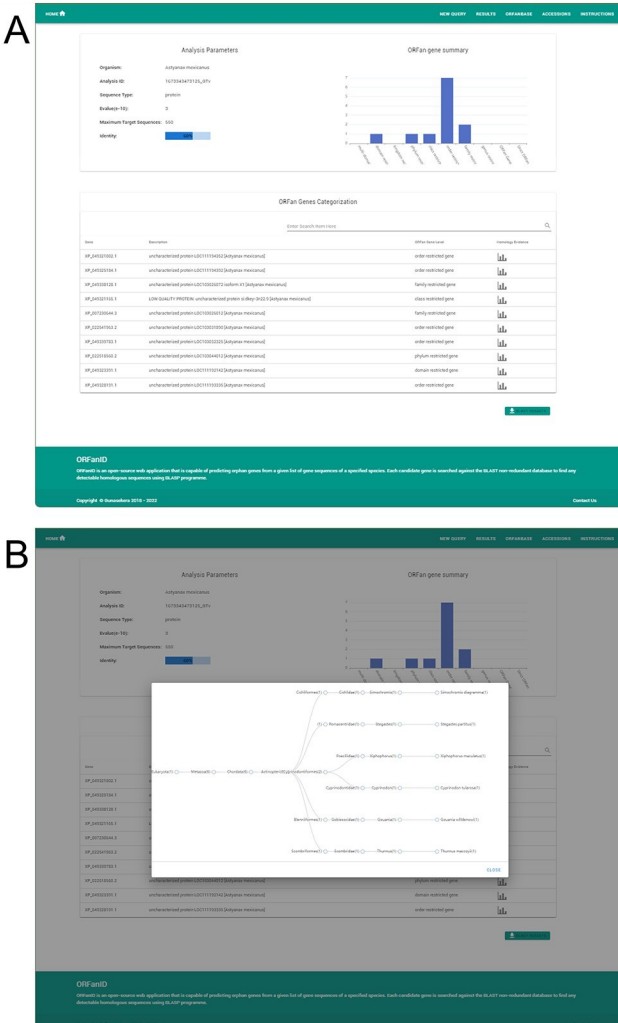

**Fig 6. ORFanID results page.** (A) This page consists of a top left table that summarizes the metadata of the analysis, a graph on the top right that depicts the taxonomy of the TRGs or orphan genes for easy visualization, and a table at the bottom that displays the categorization levels and description of the analyzed genes. (B) ORFanID BLAST Results in Graphical Display. The homology results can be viewed graphically by selecting any of the graph icons at the bottom of the results page (Fig 6A). This interactive chart visualizes the number of BLAST matches for each taxonomic rank. Each node in the tree represents the related orthologs found at each taxonomic rank, and. the graphical display is expandable and collapsible by taxonomy level. In addition, all the tables on the results page are sortable by column and can be easily filtered using the fuzzy search functionality. In addition to its web service, ORFanID can be downloaded and installed on a local server, but the local BLAST software must be downloaded from NCBI and run on the server. The Source-code and installation instructions are freely available online.

accurately grouped as strict species-level orphans by ORFanID. In *Saccharomyces cerevisiae*, the *de novo* genes *bsc4* and *fyv5*, which regulate DNA repair and vegetative growth, respectively, were accurately predicted as species-specific by ORFanID. Finally, we analyzed the *C. elegans*-specific gene *ify-1* required for proper chromosome segregation during cell division. ORFanID correctly categorized *ify-1* as a species-specific protein, a result confirmed by the "wormbase.org" database, which did not show any orthologous of the *ify-1* gene [34]. Taken together, these results suggest that ORFanID works accurately and reliably in classifying and identifying species-specific orphan genes. A complete list of taxonomically restricted or species-specific orphans we tested using ORFanID is available in the Supplemental Information.

**Table 1. Table showing the ORFanID classification of different genes.** Five known sequences are accurately classified by ORFanID.

| Species | Taxonomy ID | Accession | ORFanID Classification |
|---|---|---|---|
| *D. melanogaster* | 7227 | NP_723266.1 | Strict ORFan [1] |
| *D. melanogaster* | 7227 | NP_726607.1 | Phylum |
| *C.elegans* | 6239 | NP_500848.2 | Order [2] |
| *C. elegans* | 6239 | NP_494931.1 | Orfan gene [3] |
| *S.cervisiae* | 4932 | NP_001335743.1 | Strict ORFan [4] |
| *S. cervisiae* | 4932 | NP_014130.1 | Strict ORFan [5] |
| *H. sapiens* | 9606 | NP_073152.2 | Genus [6] |
| *H. sapiens* | 9606 | NP_001316897.1 | Order [7] |
| *D. rerio* | 7955 | XP_002665008.4 | ORFan gene [8] |
| *D. rerio* | 7955 | NP_571379.1 | Class [9] |

[1] Orphan gene based on Levine et al., 2006 [37]. No orthologs in orthodb.

[2] TRG based on Verster et al., 2017 [38]. WormBase & orthodb show Nematode orthologs

[3] TRG based on Verster et al., 2017 [38]. WormBase & orthodb show orthologs in *Caenorhabditis genus*

[4] *de novo* gene. No orthologs in other databases

[5] Strict Orphan based on publication (Cai J et al., 2008) [39]. BSC4 may be involved in the DNA repair pathway during the stationary phase of S. cerevisiae and contribute to the robustness of *S. cerevisiae* when shifted to a nutrient-poor environment

[6] Order restricted based on a publication and family based on ortho database

[7] Order restricted based on a publication. Orthodb shows conservation in eutheria clade

[8] It was shown to be genus restricted. According to orthodb, this protein has orthologs in the Actinopterygii class

[9] Should be present in all seeing animals. should be multi-domain

It is important to note that as the NCBI and other sequence-based databases grow, what is currently classified as an orphan may not be an orphan with the availability of new sequence information. Furthermore, the database of orphans and TRGs are dynamic and can change as sampling grows. However, we have noticed that the percentage of orphans in every species tends to stabilize with the increasing number of genomes synthesized. The ORFanID is updated regularly so that the databases and files obtained from NCBI are current.

## 3.1 ORFanID yields accurate results in comparison with other classifiers

We compared ORFanFinder to ORFanID, from five different organisms (*Caenorhabditis elegans*, *Escherichia coli*, *Homo sapiens*, *Oryza sativa*, and *Zea mays*). The results showed that ORFanID is more sensitive and gives more accurate results in classifying orphan and strict-orphan genes. It should be noted that ORFanFinder is no longer updated, and the last version of its database is from 2018. We also compared ORFanID with abSENSE and found that ORFanID accurately classifies 29% of the fungal genes as orphans and 37% of the Insect genes (S2 Table) [6]. This supports the accuracy of ORFanID in classifying orphans and TRGs. However, it should be noted that as more organisms are sequenced and their DNA becomes available in public databases, a percentage of these classified orphans may prove not to be true orphans.

## 4. Discussion

Here we present the implementation of a web-based, easy-to-use, intuitive tool designed to help geneticists, biologists, bioinformaticians, and computational biologists, identify and discover orphan genes across any taxonomic lineage. The search engine is designed to be user-friendly and requires minimal computational skills when compared to other bioinformatics

programs. The tool is based on the BLAST algorithm and uses the genomic sequences of various species made available through NCBI [1, 16]. Although previous stand-alone programs such as ORFanFinder existed in the past, their scope was limited to protein databases. ORFanID, on the other hand, is the first web-based, easy-to-use tool that allows users to identify orphan genes and TRGs using both nucleotide and protein sequences and classify any taxon of interest found in the NCBI databases.

With the increase in interest in orphan genes, researchers are using various approaches to identify and understand these unique genes with specific biological significance. Recently a new deep learning model (CNN + Transformer) was used to identify orphan genes in moso bamboo [35]. Another recently published website, eggNOG 6.0 database, provides a large number of species with functional annotations that allows the comparison of genes and potentially further our understanding of orphan genes [36]. One potential limitation of using BLAST as the approach to identify ORFan genes is that BLAST may not be able to identify gene lineages lacking detectable homologs. This could lead to the potential erroneous identification of ORFan genes. A recent study showed that many lineage-specific ORFan genes could be explained by homology detection failure [6]. However, these findings will need to be explored further, and ORFanID would allow the comparison and evaluation of many such genes. Another possible limitation is potential contamination which is a concern in using the NCBI databases. Although our program cannot directly control this concern, the NCBI databases have quality checks, manual Curation, and User Feedback redundancy to mitigate the effects of contamination.

In addition to its interactive user interface, ORFanID allows users to seamlessly use the BLAST database to investigate and categorically identify orphan genes. These findings can lead to the creation of databases of these clandestine genes that have been pre-identified at the various taxonomic levels by ORFanID, resulting in a deeper understanding of the purpose of orphan genes, their function in genomes, and their potential impact on life.

## Supporting information

**S1 Table. Test results of ORFanID functionality.**
(PDF)

**S2 Table. abSENSE vs. ORFanID comparison.**
(PDF)

**S1 File. ORFanID web-based search engine to identify orphan and taxonomically restricted genes.**
(PDF)

## Acknowledgments

We acknowledge the programming work of intern Savidu Dias, video creations by intern Megan Rupp, and research fellow Jesse Gentile, including technical support of engineering interns Kenneth Chen and Daniel Dirksen. We thank Marisa Arthur for her expert editorial assistance.

## Author Contributions

**Conceptualization:** Richard S. Gunasekera, Paul A. Nelson.

**Data curation:** Vinodh Gunasekera.

**Formal analysis:** Komal K. B. Raja, Emanuel Tundrea, Vinodh Gunasekera.

**Software:** Suresh Hewapathirana, Vinodh Gunasekera.

**Writing – original draft:** Richard S. Gunasekera, Komal K. B. Raja, Suresh Hewapathirana, Emanuel Tundrea, Vinodh Gunasekera, Thushara Galbadage, Paul A. Nelson.

**Writing – review & editing:** Richard S. Gunasekera, Komal K. B. Raja, Emanuel Tundrea, Vinodh Gunasekera, Thushara Galbadage, Paul A. Nelson.

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
