## [Decision Letter · Decision Letter 0]

11 Jul 2023

PONE-D-23-08315ORFanID: A Web-Based Search Engine for the Discovery and Identification of Orphan and Taxonomically Restricted GenesPLOS ONE

Dear Dr. Gunasekera,

Thank you for submitting your manuscript to PLOS ONE. After careful consideration, we feel that it has merit but does not fully meet PLOS ONE’s publication criteria as it currently stands. Therefore, we invite you to submit a revised version of the manuscript that addresses the points raised during the review process.

 Both reviewers are impressed by ORFanID and see its valuable contribution to the field. However, they both raise minor points which you will need to address prior to publication, in particular in relation to the definitions used in the manuscript. Both reviewers have also suggested practical changes to the tool itself. While these are good suggestions, only those comments relating to the manuscript content need to be addressed for publication in PLOS ONE.

We look forward to receiving your revised manuscript.

Kind regards,

Katherine James, Ph.D.

Academic Editor

PLOS ONE

Journal Requirements:

“The support of the Roth Family for the interns and the expert technical work of Chesalon USA, Inc. is greatly appreciated.”

“This work was supported in part by the Roth Family- received by RG. The funders did not and will not have a role in study design, data collection and analysis, decision to publish, or preparation of the manuscript.”

4. We note you have included a table to which you do not refer in the text of your manuscript. Please ensure that you refer to Table 1 in your text; if accepted, production will need this reference to link the reader to the Table.

Reviewers' comments:

Reviewer's Responses to Questions

**Comments to the Author**

1. Does the manuscript report a protocol which is of utility to the research community and adds value to the published literature?

Reviewer #1: Yes

Reviewer #2: Yes

2. Has the protocol been described in sufficient detail?

To answer this question, please click the link to protocols.io in the Materials and Methods section of the manuscript (if a link has been provided) or consult the step-by-step protocol in the Supporting Information files.

The step-by-step protocol should contain sufficient detail for another researcher to be able to reproduce all experiments and analyses.

Reviewer #1: Yes

Reviewer #2: Yes

3. Does the protocol describe a validated method?

Reviewer #1: Yes

Reviewer #2: Yes

4. If the manuscript contains new data, have the authors made this data fully available?

Reviewer #1: N/A

Reviewer #2: N/A

**5. Is the article presented in an intelligible fashion and written in standard English?**

Reviewer #1: Yes

Reviewer #2: Yes

6. Review Comments to the Author

Reviewer #1: I find this tool practically useful and straightforward to use. The website is well-documented and smooth; the YouTube video tutorial and screen caps on the website are a nice touch. This is a nice contribution to the field and will make these types of homology searches easier for users. The manuscript is straightforwardly written and concise.

Changes on two minor scientific points are in my opinion all that is necessary for publication:

1. Why is there a minimum percent identity cutoff? This is not standard in homology search. Why is this needed when there’s an E-value cutoff? It’s fine to have this, but I don’t think it appropriate to have the default set as high as 60 (per the video tutorial); many homologs are less identical than this, and the E-value is sufficient to exclude false positive here. I would like to see this either justified or removed.

2. For the sake of reproducibility, it’s important to add a note somewhere in the saved or downloaded results regarding what version of the database(s) were queried, as well as the search parameters.

I also found two practical issues with use of the website. I would recommend changes here, but leave it to the discretion of the authors.

1. I initially found the website confusing, even after watching the video tutorial and reading the manuscript. I was quite thrown by the fact that, after a search is submitted, it seems that the page to which one is taken displays all of the searches run by any user. This is a little visually overwhelming and confusing; for several minutes, I thought that the rest of the searches shown were somehow subcomponent searches of my search, because I’ve never seen a search tool that displays a complete public history in this manner. I would recommend that this be changed for clarity.

2. I take it that the “organism” field on the query page means the organism from which the query gene is taken. It would help to make this explicit (users of BLAST may confuse it with the field on the BLAST GUI that is in a similar location and named similarly but that indicates the organism or taxon in which the user is seeking to search, which is very different!).

Reviewer #2: Gunasekera et al. present a web-based program for identifying taxonomically restricted genes. The user can submit a gene to the program and obtain the smallest taxa to which it is restricted, as well as supporting information. This program addresses a real problem, the method is generally sound, and it serves a need that no other program is currently solving. I tried using the program and it seems to work well. This work is a useful contribution that should reduce the barriers to research into orphan genes. I have a few concerns and suggestions.

My first concern is with the definition of orphan gene used. The authors write: “By definition, orphan genes have no orthologous sequences in any other species” (line 53). In some other papers, orphan genes are defined as genes that lack detectable homology in other species. The distinction between a definition of orphans as lacking homologs vs. lacking detectable homologs is important. Since the program described in this paper only detects detectable homologs, it seems the second definition is a better fit. If the authors define orphans as those that lack any homologs, then the program will often simply be wrong, falsely declaring an ORF to lack homologs when in fact it does have homologs that have diverged beyond recognition. The authors do acknowledge this (line 301). But it seems to me the program can be made to be correct simply by using a common alternative definition in which orphan/taxonomically restricted gene are defined based on detectable homology rather than actual homology.

Relatedly, I would suggest both this manuscript and the program remain neutral in its presentation between the different processes that could generate orphan/taxonomically restricted genes under the “detectability” definition. Divergence beyond recognition, de novo gene birth, and horizontal transfer can all lead to genes that lack detectable homologs outside of a given taxa. The program as described cannot distinguish these hypotheses, and all are interesting, so make it clear to the reader/user that all are possible and it is up to the user to decide which is most plausible in their case. If the authors agree, I would suggest stating this clearly in the introduction.

Additional comments:

1) I don’t understand the sentence on lines 70-73. Please rephrase more clearly.

2) Why use BLASTP and PSI-BLAST but not TBLASTN? I think TBLASTN is important to include because the completeness of annotation varies between species so there may be homologs that are not annotated in some genomes.

3) Are the authors concerned about contamination? It is definitely still a problem in NCBI databases—some recent papers can be found in a search. I don’t see any defense against contamination mentioned in the manuscript, and it can lead to false inferences of homology in BLAST searches. (This is less of a problem if the only goal is constructing a conservative list of TRGs for a given taxa, but a big problem if the user also cares that the searches that assign ORFs outside the taxa are accurate).

4) Similarly, a 10-3 e-value threshold is appropriate for constructing a conservative list of orphan genes, but if the user also cares about accurate identification of non-orphans—if, for example, the goal is to compare genes in a narrow taxonomic range to those of broader taxonomic range—then a stricter threshold should probably be used. It is good that the program gives a choice of threshold, and I would suggest mentioning this point so users are aware of the considerations.

5) The suggested application to species identification of DNA on lines 307-314 seems implausible to me—there is enough species specificity in taxonomically widespread genes that orphans do not seem to provide much advantage for species identification. The authors don’t give any citations to suggest that this is a real problem.

7. PLOS authors have the option to publish the peer review history of their article (what does this mean?). If published, this will include your full peer review and any attached files.

Reviewer #1: No

Reviewer #2: No

---

## [Author Response · Author response to Decision Letter 0]

16 Aug 2023

The Response to Reviewers document is attached as a separate PDF file with the resubmission document labeled "Response to Reviewers".

---

## [Decision Letter · Decision Letter 1]

25 Aug 2023

ORFanID: A Web-Based Search Engine for the Discovery and Identification of Orphan and Taxonomically Restricted Genes

PONE-D-23-08315R1

Dear Dr. Gunasekera,

We’re pleased to inform you that your manuscript has been judged scientifically suitable for publication and will be formally accepted for publication once it meets all outstanding technical requirements.

Kind regards,

Katherine James, Ph.D.

Academic Editor

PLOS ONE

Additional Editor Comments (optional):

Reviewers' comments:

Reviewer's Responses to Questions

**Comments to the Author**

1. Does the manuscript report a protocol which is of utility to the research community and adds value to the published literature?

Reviewer #1: Yes

Reviewer #2: Yes

2. Has the protocol been described in sufficient detail?

To answer this question, please click the link to protocols.io in the Materials and Methods section of the manuscript (if a link has been provided) or consult the step-by-step protocol in the Supporting Information files.

The step-by-step protocol should contain sufficient detail for another researcher to be able to reproduce all experiments and analyses.

Reviewer #1: Yes

Reviewer #2: Yes

3. Does the protocol describe a validated method?

Reviewer #1: Yes

Reviewer #2: Yes

4. If the manuscript contains new data, have the authors made this data fully available?

Reviewer #1: Yes

Reviewer #2: N/A

**5. Is the article presented in an intelligible fashion and written in standard English?**

Reviewer #1: Yes

Reviewer #2: Yes

6. Review Comments to the Author

Reviewer #1: If the authors make the changes to the code/website that they have proposed in their response, I am satisfied with the changes and support publication of the manuscript.

Reviewer #2: All my concerns have been addressed in the revision. The protocol serves a useful need and is well explained in this manuscript.

7. PLOS authors have the option to publish the peer review history of their article (what does this mean?). If published, this will include your full peer review and any attached files.

Reviewer #1: No

Reviewer #2: No

---

## [Editor Report · Acceptance letter]

26 Sep 2023

PONE-D-23-08315R1 

ORFanID: A Web-Based Search Engine for the Discovery and Identification of Orphan and Taxonomically Restricted Genes 

Dear Dr. Gunasekera:

I'm pleased to inform you that your manuscript has been deemed suitable for publication in PLOS ONE. Congratulations! Your manuscript is now with our production department. 

Kind regards, 

on behalf of

Dr. Katherine James 

Academic Editor

PLOS ONE